# The Constitutive Activity of Spinal 5-HT_6_ Receptors Contributes to Diabetic Neuropathic Pain in Rats

**DOI:** 10.3390/biom13020364

**Published:** 2023-02-15

**Authors:** Nazarine Mokhtar, Marcin Drop, Florian Jacquot, Sylvain Lamoine, Eric Chapuy, Laetitia Prival, Youssef Aissouni, Vittorio Canale, Frédéric Lamaty, Paweł Zajdel, Philippe Marin, Stéphane Doly, Christine Courteix

**Affiliations:** 1Université Clermont Auvergne, INSERM, NEURO-DOL, 63000 Clermont-Ferrand, France; 2IBMM, Université de Montpellier, CNRS, INSERM, 34094 Montpellier, France; 3Faculty of Pharmacy, Jagiellonian University Medical College, 9 Medyczna Str., 30-688 Kraków, Poland; 4Institut de Génomique Fonctionnelle, Université de Montpellier, CNRS, INSERM, 34094 Montpellier, France

**Keywords:** neuropathic pain, diabetes, 5-HT_6_ receptor, constitutive activity, inverse agonism, mTOR, Streptozocin

## Abstract

Diabetic neuropathy is often associated with chronic pain. Serotonin type 6 (5-HT_6_) receptor ligands, particularly inverse agonists, have strong analgesic potential and may be new candidates for treating diabetic neuropathic pain and associated co-morbid cognitive deficits. The current study addressed the involvement of 5-HT_6_ receptor constitutive activity and mTOR signaling in an experimental model of diabetic neuropathic pain induced by streptozocin (STZ) injection in the rat. Here, we show that mechanical hyperalgesia and associated cognitive deficits are suppressed by the administration of 5-HT_6_ receptor inverse agonists or rapamycin. The 5-HT_6_ receptor ligands also reduced tactile allodynia in traumatic and toxic neuropathic pain induced by spinal nerve ligation and oxaliplatin injection. Furthermore, both painful and co-morbid cognitive symptoms in diabetic rats are reduced by intrathecal delivery of a cell-penetrating peptide that disrupts 5-HT_6_ receptor-mTOR physical interaction. These findings demonstrate the deleterious influence of the constitutive activity of spinal 5-HT_6_ receptors upon painful and cognitive symptoms in diabetic neuropathic pains of different etiologies. They suggest that targeting the constitutive activity of 5-HT_6_ receptors with inverse agonists or disrupting the 5-HT_6_ receptor-mTOR interaction might be valuable strategies for the alleviation of diabetic neuropathic pain and cognitive co-morbidities.

## 1. Introduction

Diabetic neuropathy (DN) represents a major complication of type 1 and type 2 diabetes mellitus (T1DM, T2DM). It is estimated to affect 50–70% of the 537 million people with diabetes worldwide [1] and is considered the most common neuropathy in the western world [2]. Clinical manifestations of diabetic neuropathy are various and mainly include diabetic distal symmetric sensory or sensorimotor neuropathy developing as a « dying-back » neuropathy. It affects the most distal extremities first then extends to the trunk and occasionally to the upper limbs corresponding to a « glove and stocking » distribution [3].

Painful diabetic neuropathy (pDN) is defined as « pain as a direct consequence of abnormalities in the peripheral somatosensory system in people with diabetes » [4,5]. In 2013, in France, patients suffering from chronic pain with neuropathic characteristics represented 20.3% of diabetic patients; this prevalence was found to be similar between patients with T1DM (14.7%) and T2DM (24.7%) [6] and tends to increase with longer duration of diabetes. Clinical signs of pDN include evoked and spontaneous pain and sensory loss. Painful DN is often associated with symptoms of anxiety and depression [7], sleep disorders [8], cognitive deficits [9], and decreased quality of life [6,10,11].

International guidelines recommend amitriptyline (tricyclic antidepressant), duloxetine (serotonin and noradrenaline reuptake inhibitor), and pregabalin or gabapentin (alpha2-delta calcium channel ligands) as first-line agents for symptomatic analgesic therapy in patients with diabetic peripheral neuropathic pain. Nevertheless, due to its severity and chronicity, neuropathic pain is difficult to manage, and first-line treatments have unsatisfying efficacy, as attested by the number of patients needed to treat (NNT) for 50% pain relief being above 3 [12]. Until now, there has been no recommendation regarding which first-line agent to use first and which alternative to use when pain relief is suboptimal. Of interest, combination treatments such as amitriptyline supplemented with pregabalin and duloxetine supplemented with pregabalin, led to improved pain relief in patients with insufficient pain control and could be new options in the management of diabetic peripheral neuropathic pain [13]. The difficulties in treating diabetic neuropathic pain remain in the complexity of pathophysiological mechanisms involved in diabetic neuropathy and in the heterogeneity of the patient’s response to analgesics [14].

The serotonin type 6 (5-HT_6_) receptor is expressed in the dorsal spinal cord in interneurons receiving non-painful mechanical inputs [15]. Through the Gαs protein, the 5-HT_6_ receptor is positively coupled to adenyl-cyclase/cyclic adenosine monophosphate (cAMP)/protein kinase A signaling, leading to the production of cAMP and neuron depolarization. In addition to this canonical pathway, the 5-HT_6_ receptor was shown to engage the mechanistic target of rapamycin (mTOR) [16] and cyclin-dependent-kinase 5 (Cdk5) pathways [17]. 5-HT_6_ receptor, like other serotonin receptors, is also able to be spontaneously active in the absence of endogenous serotonin [18]. Constitutive activity of 5-HT_6_ receptors has been reported for recombinant and native receptors [17,19] and in vivo [20,21].

Preclinical studies performed in animal models of neuropathic pain resembling human neuropathic pain with a high translational value have shown that 5-HT_6_ receptor ligands have a strong analgesic potential and may be new candidates to treat traumatic and chemotherapy-induced peripheral neuropathy (CIPN) [21,22]. The mechanism of action of such ligands relies on their ability to abolish the constitutive activity, serotonin independent, of 5-HT_6_ receptors, abnormally enhanced in these pathological situations. The drugs acting as inverse agonists were shown to inhibit the mechanistic target of rapamycin (mTOR) signaling upon 5-HT_6_ receptor dependence in the dorsal horn spinal cord, whereas 5-HT_6_ receptor-neutral antagonists failed to inhibit mTOR activity and alleviate neuropathic pain [21]. PZ-1386 and PZ-1179 belong to a class of 2-phenyl-1H-pyrrole-3-carboxamides and behave as inverse agonists of 5-HT_6_ receptor with Ki values of 23 nM and 30 nM, respectively [22,23]. Pharmacokinetic studies revealed that both compounds could pass through the blood-brain barrier reaching Cmax in the brain after 30 min (PZ-1386) and 5 min (PZ-1179). In addition to these favorable features, PZ-1386 was shown to reduce tactile allodynia in spinal nerve ligation (SNL)-induced traumatic neuropathic pain [23], and PZ-1179 was shown to exhibit procognitive properties in rats [22].

Since diabetes is a systemic disease impacting numerous metabolic pathways, including mTOR signaling, we hypothesized that targeting the 5-HT_6_-mTOR pathway may have considerable effects on painful diabetic neuropathy. Indeed, it is well established that mTOR signaling plays an important role in glucose homeostasis by regulating pancreatic β-cell function [24] and neuronal functions. The experimental model of streptozocin (STZ)-induced diabetic rats is currently used to study the pathophysiology of diabetic neuropathy; its complications include pain [25,26] and associated co-morbid cognitive deficit [27]. In this model, a decreased expression in the dorsal horn spinal cord of the adaptor protein APPLI, critical for insulin signaling, has been observed. This is accompanied by an increased p-mTOR expression and an exacerbation of hyperalgesia [28], suggesting that mTOR inhibition would be effective in relieving diabetic neuropathic pain. Accordingly, metformin, an indirect AMPK activator, which negatively regulates mTOR activity, was shown to prevent and reverse neuropathic pain in the spared nerve injury model in mice [29]. This effect was associated with decreased microglia activation in dorsal horn spinal cord. Metformin was also shown to attenuate neuropathic pain via AMPK/NF-kB signaling in dorsal root ganglia of diabetic rats [30]. These data, reporting the beneficial effect of mTOR inhibition in the dorsal horn spinal cord or the dorsal root ganglia, together with our previous observations reporting the anti-allodynic effect of mTOR inhibition in CIPN and traumatic neuropathic pain models [21,22], prompted us to examine whether modulating 5-HT_6_ receptor activity and signaling in the spinal cord (i.e., where the pain signal is highly modulated by central top-down controls), could be effective in painful diabetic neuropathy. For this, we investigated the involvement of 5-HT_6_ receptor constitutive activity and mTOR signaling in diabetic neuropathic pain and co-morbid cognitive symptoms induced by STZ administration in the rat using newly developed 5-HT_6_ receptor inverse agonists, PZ -1386, and PZ-1179, two 2-phenyl-1*H*-pyrrole-3-carboxamide derivatives of PZ-1388. In STZ diabetic rats, we show that blocking the constitutive activity of spinal 5-HT_6_ receptors reduces mechanical hyperalgesia and co-morbid cognitive symptoms. The 5-HT_6_ receptor ligands also reduced tactile allodynia in traumatic and toxic neuropathic pain induced by SNL and oxaliplatin (OXA) injection. mTOR inhibition, as well as disruption of 5-HT_6_-mTOR interaction, also attenuated diabetic neuropathic pain. These results suggest that STZ-induced diabetes causes dysregulation of the 5-HT_6_ signaling pathway and that inverse agonists could be a reliable strategy to alleviate diabetic neuropathic pain and associated cognitive co-morbidity.

## 2. Materials and Methods

### 2.1. Ethics

All the experiments described here on animals were conducted in accordance with the ARRIVE guidelines [31] and were reviewed and approved by the local Animal Experiment Ethics Committee of Auvergne (CEMEA) and by the French Ministry of Higher Education, Research and Innovation (authorizations *n*° 5175 (STZ rats), *n*° 17573 (SNL rats) and *n*° 24498 (OXA rats)). All efforts were made to minimize the suffering and the number of animals used.

### 2.2. Animals and Models of Neuropathic Pain

Sprague–Dawley male rats (initial weight 200–250 g) were purchased from Janvier Labs (Le Genest-Saint-Isle, France) one week before starting the experiment. Rats were housed, three (diabetic rats) or four (healthy rats) in a cage under standard laboratory conditions with a 12 h light/dark cycle, stable temperature (22 ± 1 °C), controlled humidity (55 ± 10%) and were given food and water ad libitum. At the end of the experiments, animals not collected for biochemical studies were euthanized by progressive carbon dioxide inhalation (10–30 %/min).

The previously described streptozocin (STZ)-induced neuropathic pain model was used [26]. Diabetes was induced by a single intraperitoneal (i.p.) injection of STZ (75 mg kg^−1^, Sigma–Aldrich, St-Quentin-Fallavier, France) dissolved in water for injections. An equal volume of water for injections was administrated (i.p.) to healthy rats to serve as a control group. Diabetes was confirmed one week after injection by measuring blood glucose levels in the tail vein with the Accu-Chek^®^Performa glucometer (Roche, France). At this stage, diabetic rats, i.e., animals with blood glucose > 14 mM, received a subcutaneous (s.c.) injection of insulin (LANTUS^®^, Sanofi-Aventis, Paris, France, 2 I.U./injection every other day) to ensure a good clinical condition during the entire experiment.

The spinal nerve ligation (SNL) model was performed as previously described [32]. Briefly, under xylazine 2%/ketamine (75 mg/kg i.p) anesthesia, left spinal nerve L5 was tightly ligated (4–0 silk suture, Mersilk, Ethicon LLC Johnson & Johnson, San Lorenzo, Puerto Rico). After surgery, animals were treated with a non-steroidal anti-inflammatory drug (Meloxicam, 2 mg/kg s.c., one injection per day for 2 days). The animals were allowed to recover and were monitored routinely to ensure good health. Signs of disability and distress were absent, and animals did not develop post-operative sensory loss or motor deficit as a consequence of surgery.

The chemotherapy-induced peripheral neuropathy (CIPN) model was induced by a single i.p. injection of oxaliplatin (OXA, 6 mg/kg) dissolved in a 5% glucose solution [33].

### 2.3. Behavioral Tests

Paw pressure test. Mechanical hyperalgesia to paw pressure was assessed using the Ugo Basile analgesia meter (Bioseb^®^, Vitrolles, France) [34]. The test consists of the application of an increasing mechanical force exerting important support to the dorsum of the rat’s left hind paw until a vocalization or a struggle response occurs (vocalization threshold expressed in arbitrary units, a.u.). The maximum force applied is limited to 450 a.u. to avoid tissue damage. The percentage of pain improvement (% PI) was calculated according to the formulae: % PI = [(Post-drug threshold − Pre-drug threshold) / (Pre-STZ threshold − Pre-drug threshold)] × 100. To investigate the global effects of the drugs, areas under the time-course curves (A.U.C) of the vocalization threshold changes between 0 and 180 min were calculated using the trapezoidal rule.

Von Frey hair test. In SNL and OXA rats, tactile hypersensitivity to light mechanical stimulation was assessed using the von Frey hair test [35]. Each rat was confined in clear Plexiglas boxes placed on an elevated metal mesh floor. Von Frey filaments were applied perpendicularly to the central plantar surface of the hind paw for 5 s in ascending order of force (1.4–26 g). Paw withdrawal or licking was considered a positive response, and the next weaker filament was applied. In case of a negative response (no paw withdrawal or licking), the next stronger filament is applied. This paradigm continued until four measurements were obtained after an initial change of behavior or until four consecutive negative responses or five consecutive positive responses were obtained. The 50% threshold (expressed in grams, g) was calculated using the Up-Down method and Dixon’s formulae [36].

Novel object recognition test. Episodic memory was evaluated using the novel object recognition (NOR) test as previously described [37]. Only STZ diabetic rats in which the vocalization threshold was reduced by 25% were included. Rats were habituated to the arena (1-m length, 1-m width) one day before the test. On the test day, rats had a 5-min familiarization session with two identical objects under dim light conditions (30 lux). Rats were transferred back to the home cage for 5 min inter-trial intervals. Then, rats were reintroduced in the arena for training sessions, with one of the original objects replaced by a novel one. The objects were plastic or wood toys (10-cm height, 7-cm length, 7-cm width) and were cleaned with 70% ethanol between sessions. The sequence of presentation and the location of the objects were randomly assigned to each rat. Exploratory behavior was defined as licking or touching the object while sniffing with active vibrissae. Novel object recognition memory was expressed as recognition index (D2 score), defined as the ratio of exploration time of novel object/total objects exploration time.

### 2.4. Chemicals

SB258585 hydrochloride (5-HT_6_ receptor antagonist, MW = 523.8 g.mol^−1^) and morphine hydrochloride were purchased from Biotechne Ltd. (Abingdon, UK).

The 5-HT_6_ receptor neutral antagonist IIQ hydrochloride, 4-{[5-methoxy-3-(1,2,3,6-tetrahydropyridin-4-yl)-1H-indol-1-yl]sulfonyl}isoquinoline dihydrochloride (MW = 2711.1 g.mol^−1^), was synthetized as previously described [38].

The 5-HT_6_ receptor inverse agonists PZ-1388 hydrochloride, 2-(3-fluorophenyl)-1-[(3-chlorophenyl)sulfonyl]-*N*-(piperidin-4-yl)-1*H*-pyrrole-3-carboxamide hydrochloride (MW = 534.86 g.mol^−1^), PZ-1386 hydrochloride, 2-(4- fluorophenyl)-1-[(3-chlorophenyl)sulfonyl]-*N*-(piperidin-4-yl)-1*H*-pyrrole-3-carboxamide (MW = 534.86 g.mol^−1^) and PZ-1179 hydrochloride, (*R*)-2-(4-fluorophenyl)-1-[(3-chlorophenyl)sulfonyl]-*N*-(pyrrolidine-3-yl)-1*H*-pyrrole-3-carboxamide hydrochloride (MW = 521.33 g.mol^−1^) were synthetized as previously described [21,22,23].

Tat-VEPE ([NH2]YGRKKRRQRRR-FFVTDSVEPVE[COOH], purity > 98%, MW = 2711.1 g.mol^−1^), and Tat-SCA ([NH2]YGRKKRRQRRR-TVNEK-VSCA[COOH], purity > 98%, MW = 2491.2 g.mol^−1^) peptides were synthetized by Thermo Fisher Scientific (Illkirch-Graffenstaden, France).

### 2.5. Cell Culture and Transfection

The HEK293 cells (ATCC) were cultured in DMEM (Gibco, Thermo Fisher Scientific) containing 10% fetal bovine serum (Gibco, Thermo Fisher Scientific) and 1% penicillin/streptomycin (Sigma) at 37 °C in a 5% CO_2_ enriched humidified atmosphere. Transfection (1 μg cDNA) was performed with a Viromer RED kit (Lipocalyx), and the transfected cells were cultured in the same growth medium for 48 h before use. The CMV-5-HT_6_-GFP plasmid was provided by Philippe Marin (Institut de Génomique Fonctionnelle, Montpellier, France)

### 2.6. Western Blot Analysis

Cultured cells were solubilized in lysis buffer (75 mM Tris, 2 mM EDTA, 12 mM MgCl_2_, 10 mM CHAPS, protease and phosphatase inhibitor mixture EDTA free (Roche cOmplete^TM^) for 2 h at 4 °C. Samples were centrifuged at 11,000 rpm for 20 min at 4 °C. Solubilized proteins were mixed with Laemmli buffer, heated at 95 °C for 5 min. Equal amounts of protein (25 µg) from each sample were resolved onto 10% acrylamide SDS-PAGE gels (TGX Stain-Free^TM^ FastCast^TM^, Biorad) (1.5 h at 90 V) in Tris-Glycine-SDS buffer. Proteins were transferred to nitrocellulose membrane (Trans-Blot^®^ Turbo^TM^, Biorad). Membranes were blocked for 1 h in Tris-buffered saline containing 0.1% Tween-20 and 5% BSA at room temperature. Then membranes were immunoblotted with primary antibodies overnight at 4 °C: anti-total mTOR (1:1000, Cell Signaling Technologies, ref: 2983S), anti-phospho-Ser^2448^-mTOR (1:1000, Cell Signaling Technologies, ref: 2976S), anti-total S6 (1:1000, Cell Signaling Technologies, ref: 2217S), anti-phospho-Ser^240/244^-S6 (1:1000, Cell Signaling Technologies, ref: 2215S), anti-α-actin (1:1000, Cell Signaling Technologies, ref: 6487S). After several washes, membranes were incubated with anti-rabbit HRP-conjugated secondary antibody (1:5000, Thermo Fisher Scientific) for 1.5 h at room temperature. Immunoreactivity was detected with an enhanced chemiluminescence method using the Chemidoc Touch imaging system (Clarity^TM^ Bio-Rad). Immunoreactive bands were quantified by densitometry using Image Lab 6.1.

### 2.7. Experimental Design

Baseline mechanical sensitivity was determined before STZ injection and again 21 days after. Similarly, baseline tactile threshold was determined before SNL surgery or OXA injection and again 14 days or 3 days after, respectively. At those times, STZ-D, SNL, and OXA rats were considered hyperalgesic (STZ-D rats) or allodynic (SNL- and OXA-rats) if their thresholds were reduced to at least 25%, compared with thresholds measured before the induction of the neuropathy. 

Then, for each experimental series, hyperalgesic/allodynic rats were randomly assigned to treatment groups. Thresholds were measured before then every 30 or 60 min for 120 or 180 min after drug injection to follow the effect of the treatments.

Vehicles SB258585, PZ-1388, PZ-1386, and PZ-1179 were injected intraperitoneally (i.p., 5 mL/kg). Rapamycin and Tat-VEPE peptide were administered intrathecally (i.t., 10 µL/rat).

For experiments exploring the involvement of spinal 5-HT_6_ receptors in the effect of inverse agonists, the neutral antagonist IIQ was administered i.t. (10 µL/rat) just before i.p. administration of the inverse agonist. Then, thresholds were measured for 120 or 180 min to monitor the effect of the treatments.

For experiments exploring the effect of drugs on cognitive dysfunction, the drugs were administered i.p., and the NOR test was performed with a delay coinciding with the maximal antihyperalgesic effect of the drug.

All the in vivo experiments were performed blind using a parallel group design. Treatments were randomized. Different animals were used in each experiment.

### 2.8. Statistical Analysis

Data were analyzed using the GraphPad Prism software (GraphPad Prism 6 Software). Biochemical and behavioral data were analyzed by a one-way ANOVA followed by Tukey’s post-hoc test or a two-way ANOVA, followed by Dunnett’s post-hoc test. Results were expressed as mean ± standard error of the mean (S.E.M). The statistical significance was set at 5% (*p* < 0.05). All statistics are detailed in Table 1 and Appendix A.

## 3. Results

### 3.1. Diabetes-Induced Mechanical Hyperalgesia

STZ was used to induce pDN. Mechanical hyperalgesia was determined by paw pressure-induced vocalization threshold. As expected [39], STZ-induced diabetes was associated with disturbances of biological and clinical parameters such as hyperglycemia, polydipsia, polyuria, polyphagia, and a halt in weight gain. Ninety-eight percent of STZ-injected rats developed hyperglycemia. Acute mortality occurring within one week of STZ injection was observed in only 2/273 STZ-injected rats. None of the hyperglycemic animals displayed weight loss greater than 10% of their initial weight, motor dysfunction, or prostration. This relatively good health status of diabetic rats is related to insulin administration (every other day). Seventy-nine percent of hyperglycemic rats showed a significant reduction in vocalization threshold 21 days after induction of diabetes (182.8 ± 2.5 a.u. vs. 318.8 ± 1.9 a.u. before STZ), suggesting mechanical hyperalgesia. Twenty-one percent of hyperglycemic rats were not hyperalgesic and were discarded from the study. 

### 3.2. Serotonin 6 Receptor Blockade Attenuated Mechanical Hyperalgesia in STZ-Induced Diabetic Rats

To explore the role of constitutive activity of 5-HT_6_ receptor in the development of pDN, two well-characterized inverse agonists, PZ-1388 and SB258585, were injected intraperitoneally (i.p.), and mechanical thresholds to paw pressure were measured up to 180 min after injection. Pharmacological inhibition of the 5-HT_6_ receptor constitutive activity by PZ-1388 and SB258585 administered at the doses of 5, and 25 µmol/kg significantly attenuated diabetes-induced mechanical hyperalgesia 60–150 or 180 and 30–180 min after injection for PZ-1388 and SB258585, respectively (Figure 1a,b, Table 1). The time course of the antihyperalgesic effect of the drugs showed a maximal effect at 90 min for PZ-1388 and 120 min for SB258585 and corresponded to a pain improvement of 83 ± 21% and 86 ± 9%, respectively. The overall effect assessed by calculation of the area under the time-course curve (A.U.C. 0–180 min) of vocalization threshold changes of the two compounds showed no significant differences between the 5 and 25 µmol doses for each (Figure 1e, Table 1).

**Table 1 biomolecules-13-00364-t001:** Detailed statistics for figures.

Figure	Analysis	Statistics (DFn, DFs)	*p* Value
Figure 1a–d	2-way RM ANOVA	*F(63, 798)* = 12.39	*p* < 0.0001
Figure 1e	1-way ANOVA	*F(9, 117)* = 24.27	*p* < 0.0001
Figure 2b,c	2-way RM ANOVA	*F(35, 273)* = 2.812	*p* < 0.0001
Figure 2d	1-way ANOVA	*F(5, 39)* = 8.252	*p* < 0.0001
Figure 3a	2-way RM ANOVA	*F(24, 160)* = 3.399	*p* < 0.0001
Figure 3b	1-way ANOVA	*F(3, 20)* = 16.11	*p* < 0.0001
Figure 4a	2-way RM ANOVA	*F(14, 119)* = 8.237	*p* < 0.0001
Figure 4b	1-way ANOVA	*F(2, 17)* = 11.23	*p* = 0.0008
Figure 5a	1-way ANOVA	*F(2, 40)* = 6.181	*p* = 0.0046
Figure 5b,c	1-way ANOVA	*F(3, 27) =* 5.497	*p* = 0.0044

The newly synthesized inverse agonists PZ-1386 and PZ-1179 significantly reduced mTOR and S6 phosphorylation in HEK cells (Appendix A) and, as expected, significantly reduced mechanical hyperalgesia in diabetic neuropathic rats (Figure 1c,d, Table 1). This antihyperalgesic effect lasted no more than 120 min for both compounds at 5 µmol and lasted at least 180 min for the dose of 25 µmol. We observed a complete abolition of mechanical hyperalgesia 90 min after administration of the inverse agonist PZ-1179 25 µmol/kg (pain improvement of 109 ± 16%) and a partial improvement after administration of PZ-1386 (pain improvement of 62 ± 8%). Despite the high activity of PZ1179 at 90 min, the overall effect of the two inverse agonists was similar, as shown by the calculation of area under the A.U.C. of vocalization threshold changes (0–180 min) (Figure 1e, Table 1).

Since the anti-allodynic effect of PZ-1388 and SB258585 has been reported to be dependent on spinal 5-HT_6_ receptors in SNL and OXA rats [21], we sought their involvement in pDN. To investigate the role of spinal 5-HT_6_ receptors in diabetes-induced hyperalgesia, the neutral antagonist IIQ (2 nmol/rat) was injected intrathecally (i.t.) just before i.p. administration of PZ-1179 (25 µmol/kg) or PZ-1386 (25 µmol/kg) or vehicle (water for injections) (Figure 2a) and vocalization thresholds to paw pressure were measured up to 180 min. Consistent with previous findings indicating that neuropathic pain results from a constitutive activity of 5-HT_6_ receptors rather than tonic 5-HT_6_ receptor activation by endogenous serotonin [21], i.t. administration of the selective neutral antagonist IIQ did not alter the paw pressure-induced vocalization threshold in STZ diabetic rats but completely abolished the antihyperalgesic effect of PZ-1179 (PZ-1179 + IIQ group) (Figure 2b, Table 1). Behavioral results obtained after co-administration of PZ-1386 and IIQ showed an inhibition of the antihyperalgesic effect of PZ-1386 (PZ-1386 + IIQ group) despite a slight elevation of vocalization thresholds to paw pressure (Figure 2c, Table 1). The A.U.C. of the vocalization thresholds changes, which confirms the loss of the antihyperalgesic effect of PZ-1179 and PZ-1386 in the presence of the neutral antagonist (Figure 2d, Table 1). These results indicate that spinal 5-HT_6_ receptors’ constitutive activity plays a critical role in pDN.

Next, we wondered whether the results obtained on diabetes-induced mechanical hyperalgesia could be extended to traumatic and toxic neuropathic pain in the SNL and CIPN models. PZ-1386 (25 µmol/kg) not only abolished tactile hypersensitivity evoked by von Frey hair stimulation in OXA rats (Appendix A) but also exerted a short antinociceptive effect 90 min post-injection (Appendix A). This brief antinociceptive effect was also observed with PZ-1179 (25 µmol/kg i.p.) 60 min post in SNL and OXA rats (Appendix A). Since PZ-1386 and PZ-1179 showed a transient antinociceptive effect in the rat models of traumatic and toxic neuropathic pain, we then explored the effect of both compounds on mechanical nociception (paw pressure test) in healthy rats. PZ-1386 and PZ-1179, given a dose of 10 µmol/kg i.p. (Appendix A) or 25 µmol/kg i.p. (Appendix A), failed to increase the paw pressure-induced vocalization threshold (Appendix A). Morphine (2 mg/kg, i.p.), a positive control, induced a significant increase in the vocalization threshold (Appendix A). Finally, the analgesic effect of both inverse agonists on tactile hypersensitivity (von Frey hair) was completely abolished by simultaneous co-administration of the neutral antagonist IIQ (Appendix A). These results suggest that the analgesic effect observed in neuropathic pain models is specific to mechanisms underlying chronic hypersensitivity rather than nociception and depends on the constitutive activity of 5-HT_6_ receptors.

### 3.3. Rapamycin and Tat-VEPE Attenuated Hyperalgesia Induced by Diabetes

Inhibiting mTOR activity or interfering with the physical interaction between the C-terminus of the 5-HT_6_ receptor and mTOR have been shown to be effective strategies for alleviating traumatic and toxic neuropathic pain [21]. Before exploring the involvement of 5-HT_6_ receptor-dependent mTOR signaling in diabetic neuropathic pain, we performed a behavioral experiment with the mTOR inhibitor rapamycin to ensure that the mTOR signaling is engaged. For this purpose, STZ diabetic neuropathic rats were injected i.t. with rapamycin (0.3, 3, and 10 nmol) or vehicle (water for injections, 10 µL/rat) and subjected to the paw pressure test. The results presented in Figure 3 indicate that rapamycin significantly and dose-dependently increases vocalization threshold to paw pressure 30–120 min (Figure 3a, Table 1). The maximum effect observed at 120 min (297.5 ± 23.4 a.u.) with rapamycin 10 nmol corresponds to a complete reversal of mechanical hyperalgesia (pain improvement of 114 ± 44%). The dose-dependent effect of rapamycin was illustrated by the A.U.C. of changes in the vocalization threshold measured between 0 and 180 min (Figure 3b, Table 1).

Using a peptidyl mimetic strategy disrupting the physical 5-HT_6_-mTOR interaction, we previously demonstrated that the spinal 5-HT_6_ receptor constitutively activates mTOR signaling to contribute to traumatic and toxic neuropathic pain in SNL and OXA rats [21]. The same strategy was used to examine whether inhibiting the physical interaction between the C-terminal domain of the receptor and mTOR would also ameliorate mechanical hyperalgesia to paw pressure in STZ diabetic rats. Intrathecal administration of Tat-VEPE (300 ng/rat i.t.), but not a control peptide (Tat-SCA, 300 ng/rat i.t.), reduced mechanical hyperalgesia 30–120 min after injection in painful STZ diabetic rats (Figure 4a, Table 1). The maximal antihyperalgesic effect (265.7 ± 7.1 a.u.) occurred 60 min after injection, resulting in a 68 ± 8% reversal of mechanical hyperalgesia. The overall antihyperalgesic effect of Tat-VEPE, assessed by the A.U.C. of threshold changes, confirmed the effect of Tat-VEPE compared with the control Tat-SCA peptide (Figure 4b, Table 1).

### 3.4. Effect of 5-HT_6_ Receptor Ligands and Tat-VEPE on Co-Morbid Cognitive Symptoms

Pro-cognitive effects of 5-HT_6_ receptor ligands have been reported not only in traumatic neuropathic pain models [21] but also in several paradigms of cognitive impairment [22,23,40,41]. Because diabetic rats, like diabetic humans [9], develop cognitive deficits, we next examined whether inverse agonists and peptidyl mimetic strategy would be effective in ameliorating cognitive deficits associated with chronic pain in hyperalgesic diabetic rats. We, therefore, tested the effect of PZ-1179 and PZ-1386 upon episodic memory (NOR task). STZ diabetic rats treated with vehicle failed to discriminate a novel object from the familiar one (Figure 5a, Table 1). Systemic (i.p.) administration of PZ-1386 (25 µmol/kg) but not PZ-1179 (25 µmol/kg) restored the episodic memory deficit elicited by pDN (Figure 5a, Table 1). Reminiscent of the pro-cognitive effect of PZ-1386, intrathecal injection of Tat-VEPE (300 ng/rat) increased novelty discrimination in STZ diabetic rats (Figure 5b, Table 1).

## 4. Discussion

In the present study, we explored the effect of 5-HT_6_ receptor inverse agonists and the potential role of mTOR in painful diabetic neuropathy. After diabetes, we found that mechanical hypersensitivity and co-morbid cognitive deficits were alleviated with inverse agonists but not with neutral antagonists, suggesting that the constitutive activity of the 5-HT_6_ receptor plays a critical role in the development of diabetic neuropathic pain. More importantly, while rapamycin suppressed diabetic-induced mechanical hyperalgesia, disrupting the interaction between spinal 5-HT_6_ receptors and mTOR that physically interacts with their C-terminal ending could also ameliorate mechanical hyperalgesia induced by diabetic neuropathic pain. In addition, the novel inverse agonists also suppressed tactile hypersensitivity to von Frey hair application in models of traumatic and toxic neuropathic pain.

The experimental model of STZ-induced diabetic rats is a reliable model of diabetic neuropathic pain with a high translational value [25,42,43,44]. To reduce mortality due to STZ acute toxicity, young rats aged 6 weeks were chosen at the time of STZ injection as previously preconized [45]. Furthermore, in order to prevent decreasing health status confounding behavioral readouts, STZ rats with confirmed hyperglycemia were treated with insulin every other day, and pharmacological experiments were performed 21 days post-STZ treatment as recommended [46].

PZ-1388 and its two derivatives, PZ-1386 and PZ-1179, are selective 5-HT_6_ receptor ligands characterized as inverse agonists [21,22,23]. The 5-HT_6_ receptor is a G protein-coupled receptor that exhibits constitutive activity in vivo [20], including in a preclinical model of traumatic neuropathy [21]. Constitutive activity of 5-HT_6_ receptors has been previously shown to be responsible for enhanced mTOR activity in the spinal cord of SNL and OXA rats and remains to be explored in the context of painful diabetic neuropathy. In the present study, we provide evidence of the involvement of 5-HT_6_ receptor constitutive activity in the animal model of STZ-induced diabetic rats. The effects of the 5-HT_6_ inverse agonists in diabetic rats were assessed on a painful symptom induced by sustained and increasing paw pressure, a procedure able to reveal mechanical hyperalgesia in 79% of diabetic rats. PZ-1388 and its two derivatives, PZ-1179 and PZ-1386, display similar antihyperalgesic effects with high efficacy, leading to an improvement in the pain of 60 to 109% and a relatively long duration of effect, at least 180 min, for all the molecules tested at a dose of 25 µmol/kg. 

Despite having different pharmacokinetic parameters for 2-phenyl-1*H*-pyrrole-3-carboxamide-derived 5-HT_6_ receptor inverse agonists, such as the brain-plasma ratio (4.44, 2.18, 0.64 for PZ-1388, PZ-1386, and PZ-1179, respectively), time to maximal brain concentration (Tmax value of 5 min for PZ-1179, 30 min for PZ-1388 and PZ-1386) and brain half-life (T_1/2_ value of 250.9 min for PZ-1179, 414.7 min for PZ-1388, and 1743 min for PZ-1386) [21,22,23], the antihyperalgesic effect of the 5-HT_6_ inverse agonists in diabetic rats was of similar magnitude and duration. Nevertheless, the brain Cmax values reported for PZ-1388 (1877 nM), PZ-1386 (876 nM), and PZ-1179 (882 nM) in rats receiving 10 mg/kg each (p. o.) were all much higher than the IC_50_ values at Gs signaling (164 nM, 164 nM, and 265 nM, for PZ-1388, PZ-1386, and PZ-1179, respectively). This suggests that at the dose of 25 µmol/kg (12.5 mg/kg), the theoretical brain Cmax values are 4 (PZ-1179), 6 (PZ-1386), and 14 (PZ-1388) times that of the IC_50_ at G_S_ signaling, explaining the absence of difference in the maximum magnitude of the antihyperalgesic activity. More interestingly, the dose of 5 µmol/kg may correspond to theoretical brain Cmax values 2.9, 1.34, and 0.83 times that of the IC_50_ values for PZ-1388, PZ-1386, and PZ-1179, respectively. This suggests that the overall activity measured by A.U.C. of PZ-1388 should be higher than that of PZ-1386 and PZ-1179. Despite an apparent overall effect (A.U.C.) greater for PZ-1388 than PZ-1386 and PZ-1179 (Figure 1e) there is no significant difference (Table 1), suggesting that the paw pressure test is probably not sensitive enough to reveal such a difference.

The effect currently observed with SB258585 on mechanical hyperalgesia appears similar in duration to those previously reported on tactile allodynia in SNL and OXA rats [21]. Intraperitoneal administration of SB258585 has been reported to reduce thermal hyperalgesia in STZ diabetic mice and to be effective for 120 min after injection [47]. The compound has also been shown to reverse formalin-induced secondary hyperalgesia in rats [48] and to decrease tactile allodynia in SNL rats [49].

Finally, the antihyperalgesic effect of 5-HT_6_ receptor inverse agonists is comparable to that reported in diabetic rats with pregabalin, one of the reference treatments in neuropathic pain, but it occurs at a lower dose (equivalent to 12.5 mg/kg) than the pregabalin dose (30 mg/kg, p. o.) [50]. Overall, these experimental observations are much more satisfactory than those reported in clinical trials, with first-line treatments offering, at best, only partial relief of neuropathic pain for one patient out of three (NNT = 3) [12,51]. 

Remarkably, the analgesic effect of the inverse agonists observed in diabetic neuropathy was also observed in neuropathy of traumatic and toxic etiologies on tactile hypersensitivity in response to von Frey filaments application. The von Frey hair test is a relevant method to explore neuropathic pain in SNL and OXA rats since 70–80% of the animals present a reduction in paw withdrawal threshold compared to only 50% of STZ diabetic rats (personal data). Time follow-up of the effect of PZ-1179 shows that its anti-allodynic activity was much greater in SNL and OXA rats than its anti-hyperalgesic effect in STZ diabetic rats, leading to a threshold higher than that measured before the induction of neuropathy. The effect of PZ-1386 also seems to depend on the etiology of neuropathy; it induced a more marked anti-allodynic effect on tactile allodynia in OXA rats than previously reported in SNL rats [22]. Importantly, the lack of antinociceptive effect of both inverse agonists in healthy conditions further supports that 5-HT_6_ receptor inhibition specifically alleviates painful symptoms characteristic of neuropathic pain. 

Antagonizing spinal 5-HT_6_ receptors with a neutral antagonist did not affect mechanical hyperalgesia or tactile hypersensitivity but suppressed inverse agonist-induced analgesia. This result indicates and confirms that mechanical hyperalgesia and tactile hypersensitivity induced by diabetic, traumatic, and toxic neuropathy depend on the constitutive activity of the 5-HT_6_ receptor rather than its activation by endogenously released serotonin. Consistent with previous data obtained in SNL rats, the suppression of the analgesic effect after intrathecal administration of the antagonist also indicated that spinally located 5-HT_6_ receptors are involved in this effect. This is supported by the presence of 5-HT_6_ receptors in the dorsal spinal cord [21], expressed in excitatory interneurons receiving signals from low threshold mechanoreceptors [15]. 

We do not provide direct evidence of the attenuation of 5-HT_6_ receptor-dependent mTOR activity by inverse agonists. Nevertheless, both (i) the *in vitro* western blot analysis of the phosphorylation of mTOR and a downstream protein, S6, showing a decreased expression after 5-HT_6_ receptor inverse agonists application and (ii) the behavioral experiment conducted with intrathecal administration of the peptide Tat-VEPE designed to disrupt 5-HT_6_ receptor-mTOR interaction, indicate that mTOR, under the dependence of the 5-HT_6_ receptor, is likely involved in diabetic neuropathy-induced mechanical hyperalgesia. Indeed, Tat-VEPE reduced mechanical hyperalgesia leading to a 68% improvement in pain. The dose-dependent effect of rapamycin appeared more marked than that of Tat-VEPE peptide (maximal effect corresponding to the total suppression of allodynia). As mTOR is a ubiquitous signaling protein, the effects observed after intrathecal injection of rapamycin are suggestive of broader effects of rapamycin in the central nervous system (CNS) (other than those dependent on the 5-HT_6_ receptor) and, in particular, effects targeting mTOR activation elicited by APPLI deficiency, a mechanism that has been shown in the dorsal horn of the spinal cord of hyperalgesic diabetic rats [52]. Taken together, these data provide evidence that the 5-HT_6_ receptor-dependent mTOR signaling pathway contributes to mechanical hyperalgesia in diabetic rats.

Although PZ-1179 has been shown to prevent scopolamine-induced cognitive deficits in the NOR test in rats [23] and is analgesic in the present study, it failed to improve episodic memory in STZ diabetic rats. This discrepancy may be due to the lower CNS penetrance and shorter brain T_1/2_ of PZ-1179 compared to PZ-1386. Further administration modalities need to be performed to support this hypothesis. Finally, consistent with recent findings showing the pro-cognitive effect of SB258585 and PZ-1388 in SNL rats [21], intraperitoneal administration of PZ-1386 and intrathecal injection of Tat-VEPE improved co-morbid cognitive deficits in diabetic rats, a highly prevalent complication of diabetes [9].

Taken together, these data suggest that 5-HT_6_ receptor inverse agonists—already in clinical development for the treatment of cognitive impairment in dementia and psychoses—and strategies disrupting 5-HT_6_ receptor-mTOR interaction, might be new therapeutic approaches for the treatment of diabetic neuropathic pain, in addition to neuropathic pain of traumatic and toxic etiologies. 

## 5. Conclusions

The present study demonstrated that the constitutive activity of the 5-HT_6_ receptor in the dorsal horn of the spinal cord plays a critical role in the pathophysiology of diabetic neuropathic pain via dependent activation of mTOR signaling, suggesting a potential therapeutic interest in targeting constitutively active 5-HT_6_ receptors or 5-HT_6_ receptor-mTOR complexes for the treatment of neuropathic pain and cognitive co-morbidities.

## Figures and Tables

**Figure 1 biomolecules-13-00364-f001:**
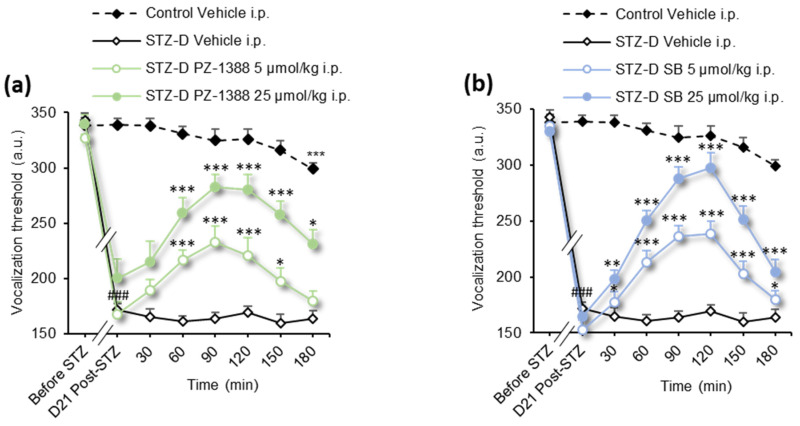
Systemic administration of 5-HT_6_ receptor inverse agonists reduces diabetic neuropathic pain (paw pressure test). (**a**) Intraperitoneal administration of PZ-1388, (**b**) SB258585, (**c**) PZ-1386, and (**d**) PZ-1179 (5 and 25 µmol/kg each) but not the vehicle (water for injections) improves mechanical hyperalgesia to paw pressure in STZ diabetic (STZ-D) rats (*n* = 11–15/group). Vocalization thresholds are expressed as arbitrary units (a.u.). ^###^
*p* < 0.001 vs. values measured before STZ; * *p* < 0.05, ** *p* < 0.01, *** *p* < 0.001 vs. values measured before the drug/vehicle (water) injection (D21 Post-STZ), 2-way ANOVA. (**e**) Area under the curve (A.U.C.) of vocalization threshold variations in STZ-D and control rats calculated by the trapezoidal rule (in a.u.. min). * *p* < 0.05, ** *p* < 0.01, *** *p* < 0.001, vs. STZ-D Vehicle group; ^#^
*p* < 0.05 ^###^
*p* < 0.001 vs. corresponding group, 1-way ANOVA.

**Figure 2 biomolecules-13-00364-f002:**
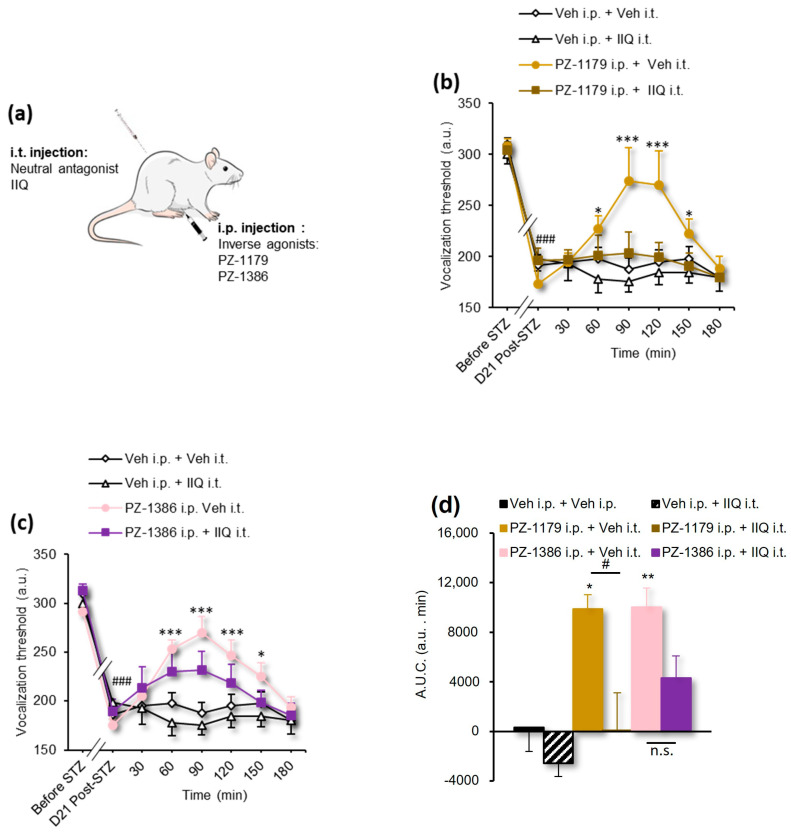
5-HT_6_ receptor constitutive activity is involved in mechanical hyperalgesia (paw pressure test) in STZ diabetic rats. (**a**) Schema illustrating the route of administration of 5-HT_6_ receptor ligands in experiment illustrated on panels b–d; (**b**) Intrathecal administration of IIQ (2 nmol/rat) but not vehicle (water for injections, 10 µL/rat) suppresses the antihyperalgesic effect of PZ-1179 (25 µmol/kg, i.p.) and (**c**) PZ-1386 (25 µmol/kg, i.p.), respectively (*n* = 6–9/group). ^###^
*p* < 0.001 vs. values measured before STZ; * *p* < 0.05, ** *p* < 0.01, *** *p* < 0.001 vs. values measured before drug/vehicle injection (D21 Post-STZ), 2-way ANOVA; (**d**) Area under the curve (A.U.C.) of vocalization threshold variations in STZ rats calculated by the trapezoidal rule (in a.u. min). * *p* < 0.05, ** *p* < 0.01, vs. Veh + Veh group; ^#^
*p* < 0.05 vs. corresponding group, 1-way ANOVA.

**Figure 3 biomolecules-13-00364-f003:**
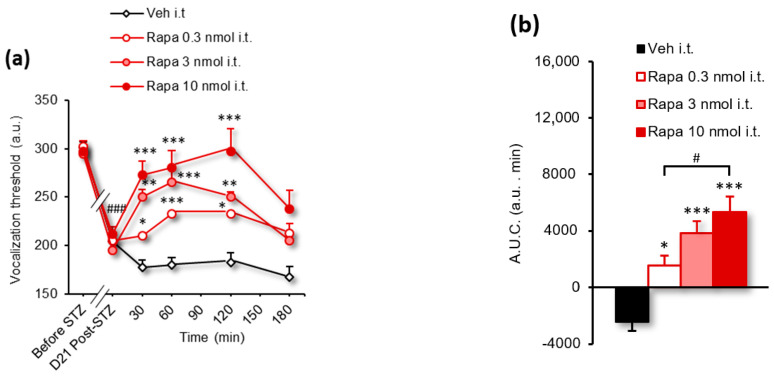
Intrathecal administration of rapamycin suppresses mechanical hyperalgesia (paw pressure test) in STZ diabetic rats. (**a**) Intrathecal administration of rapamycin (Rapa, 0.3–10 nmol/rat) but not the vehicle (Veh, water for injections) dose-dependently increases the vocalization threshold to paw pressure in STZ rats (*n* = 6/group). ^###^
*p* < 0.001 vs. values measured before STZ. * *p* < 0.05, ** *p* < 0.01, *** *p* < 0.001 vs. values measured before the drug/vehicle injection (D21 Post-STZ), 2-way ANOVA; (**b**) Area under the curve (A.U.C.) of vocalization threshold variations in STZ rats calculated by the trapezoidal rule (in a.u. min). * *p* < 0.05, *** *p* < 0.001, vs. Vehicle group; ^#^
*p* < 0.05 vs. corresponding group, 1-way ANOVA.

**Figure 4 biomolecules-13-00364-f004:**
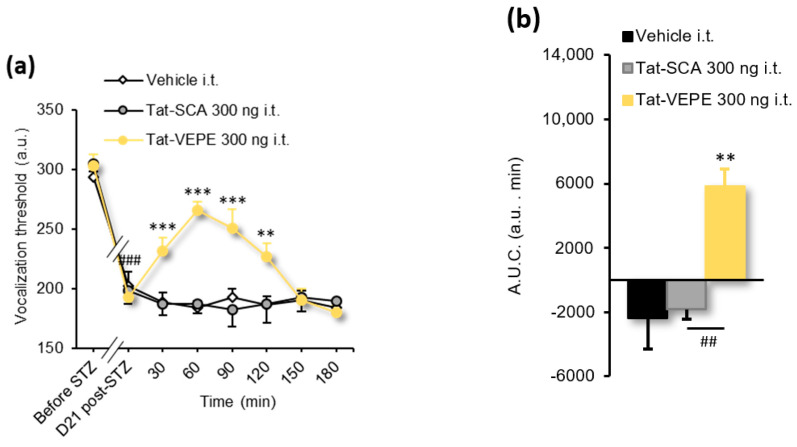
Intrathecal administration of Tat-VEPE attenuates mechanical hyperalgesia (paw pressure test) in STZ diabetic rats. (**a**) Intrathecal administration of a cell-permeable interfering peptide fused to the transduction domain of the HIV Tat protein (Tat-VEPE, 300 ng/rat) increases the vocalization threshold to paw pressure application in STZ-D rats (*n* = 6–7 /group). ^###^
*p* < 0.001 vs. values measured before STZ. ** *p* < 0.01, *** *p* < 0.001 vs. values measured before the drug/vehicle injection (D21 Post-STZ), 2-way ANOVA; (**b**) Area under the curve (A.U.C.) of vocalization threshold variations in STZ rats calculated by the trapezoidal rule (in a.u. min). ** *p* < 0.01 vs. Vehicle group; ^##^
*p* < 0.01 vs. corresponding group, 1-way ANOVA.

**Figure 5 biomolecules-13-00364-f005:**
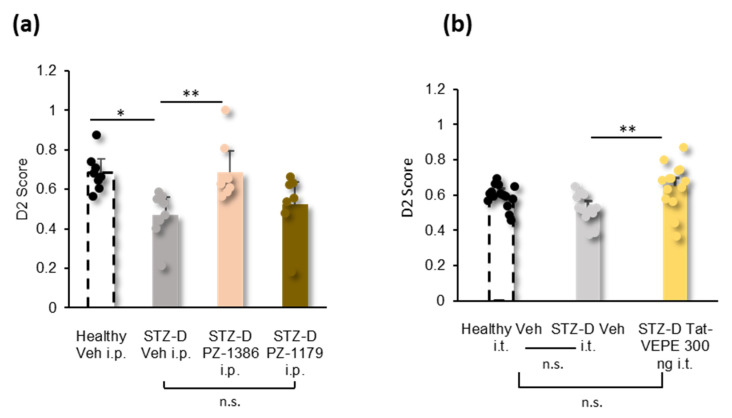
Effect of blocking the constitutive activity of 5-HT_6_ receptor or disrupting 5-HT_6_-mTOR interaction on cognitive deficits (novel object recognition test) induced by diabetic neuropathic pain in rats. (**a**) Intraperitoneal injection of PZ-1386 (25 µmol/kg) but not PZ-1179 (25 µmol/kg), or vehicle (Veh, water for injections), and (**b**) intrathecal injection of Tat-VEPE (300 ng/rat), improves novelty discrimination in STZ diabetic (STZ-D) rats (*n* = 7–16/group). * *p* < 0.05, ** *p* < 0.01, vs. STZ-D Vehicle group, 1-way ANOVA.

## Data Availability

All data supporting reported results in this article can be obtained from the corresponding author on request.

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
