# Peer review of "The Constitutive Activity of Spinal 5-HT6 Receptors Contributes to Diabetic Neuropathic Pain in Rats"

_biomolecules, 2023, doi:10.3390/biom13020364_

Round 1
Reviewer 1 Report
The main aim of this study was to test the analgesic effects of inverse agonists of serotonin type 6 (5-HT6) receptor ligands on induced neuropathic pain in rats. Three animal models of neuropathic pain were used: (1) streptozocin- (STZ) diabetic, (2) spinal nerve ligation and (3) chemotherapy induced neuropathy. The results showed that neuropathic manifestations are suppressed by the administration of 5-HT6 receptor inverse agonists.
Although the results of this study might promise a new therapeutic strategy in treating diabetic neuropathic pain as well as neuropathic pain following nerve injury or chemotherapy treatment, these are my concerns and comments:
· In the title: authors state that “spinal 5-HT6 receptors contributes……”. The results of this study showed that the spinal 5-HT6 receptors do not contribute, they rather attenuate neuropathic manifestations!
· Allodynia in human is pain due to a stimulus that does not normally elicit pain. This cannot be accurately tested in rodents. Therefore, in my opinion, what the authors referred to as allodynia or anti-allodynic should be better termed as a mechanical hypersensitivity which very likely represents mechanical hyperalgesia.
· In the introduction, authors state that the clinical signs of painful diabetic neuropathy (pDN) include evoked and spontaneous pain and sensory loss. Neither spontaneous pain nor sensory loss were investigated in the diabetic rats in this study!
· What is the current treatment of painful diabetic neuropathy in human? Why is it not effective in patients? Why an alternative or a better treatment are needed? This was not specifically stated in the introduction or discussion sections.
· Authors used different tests to assess hypersensitivity/hyperalgesia in different models of neuropathic pain. They used paw pressure test in diabetic rats and Von Frey hair test in SNL and OXA rats. Why?
· Thermal nociception was not tested in this study!
· Only STZ-diabetic rats in which the vocalization threshold was reduced by 25 % were included. What is the percentage of animals that were excluded in this study?
· In this study an intraperitoneal injection of 75 mg/Kg of STZ was used to induce diabetes and consequently diabetic neuropathic pain. This is a high dose of STZ which would result in a poor health of the animals with a high mortality rate! Also, did the animals show inactivation of motor activity, any abnormal behavior, polyuria, GIT dysfunction and weight loss?
The acute STZ toxicity and the mortality rate of the animals should be included in the results section.
· In what vehicle was the STZ prepared?
· I have noticed an inappropriate use of control experiment in this study. The experimental animals were injected with STZ and then with insulin every other day. However, animals in control groups were injected only with water!
· In general, the results are a bit confusing and should be rewritten. For example, two methods were used to assess the mechanical hyperalgesia namely, paw pressure test and Von Frey hair test. This should be clearly stated in the results and the legends of the figures!
· On lines 308-310, the results showed that PZ-1386 and PZ-1179 did not only abolish tactile allodynia in SNL and OXA rats respectively but produced significant suppression of nociceptive behavior compared to the basal level before the treatment (Figure A2). This should be reported and explained.
· Control experiments to compare the efficacy of serotonin 6 receptor blockade with the first-line therapy for diabetic neuropathy e.g., using pregabalin, or serotonin reuptake inhibitor would have been of advantage.
· In Figure A1: It looks that there is a non-specific band in the Western blot of p-mTOR which is also thicker in vehicle group compared with PZ-1179 and PZ-1386 groups!
It will better if loading control is included in the Western blot results.
· The specificity of the antibodies used in the Western blot should be discussed and added.
Reviewer 2 Report
Authors explored the effect of 5-HT6 receptor inverse agonists and the potential role of mTOR in painful diabetic neuropathy. Antagonizing spinal 5-HT6 receptors with a neutral antagonist did not affect mechanical hyperalgesia but suppressed inverse agonist-induced anti-hyperalgesia. This result indicates that mechanical hyperalgesia induced by diabetic neuropathy depends on the constitutive activity of the 5-HT6 receptor rather than its activation by endogenously released serotonin. Besides 5-HT6 receptor inverse agonists, the mechanical hyperalgesia and associated cognitive deficits are suppressed by the administration of rapamycin, possibly disrupting the interaction between spinal 5-HT6 receptors and mTOR that physically interacts with their C-terminal ending. In addition, the novel inverse agonists also reduced neuropathic pain (tactile allodynia) in models of traumatic and toxic neuropathic pain.
Despite different pharmacokinetic parameters for 2-phenyl-1H-pyrrole-3-carboxamide-derived 5-HT6 receptor inverse agonists (brain-plasma ratio, time to maximal brain concentration and brain half-life), the antihyperalgesic effect of the 5-HT6 inverse agonists in diabetic rats was of similar magnitude and duration. Authors are invited to detail discussion of these phenomena.
Author Response
In order to attempt to discuss this result, we analyzed previously published pharmacokinetic and pharmacodynamic data after 10 mg/kg p. o. of the compounds ([1,2], with all the limitations of this approach. It results that, at the dose of 25 µmol/kg, the theoretical brain Cmax values are 4 (PZ-1179), 6 (PZ-1386) and 14 (PZ-1388) times that of the IC50 at GS signaling, explaining the absence of difference in the maximum magnitude of the antihyperalgesic activity. More interestingly, the dose of 5 µmol/kg corresponds to brain Cmax value 2.9, 1.34 and 0.83 times the IC50 value for PZ-1388, PZ-1386 and PZ-1179, respectively. This suggests that the overall activity measured by A.U.C. of PZ-1388 should be higher than that of PZ-1386 and PZ-1179. Despite an apparently greater overall effect (A.U.C.) for PZ-1388 than for PZ-1386 and PZ-1179 (Figure 1e), no significant difference was found, suggesting that paw-pressure test is not sensitive enough to reveal such a difference. (l. 464-476)
- Drop, M.; Jacquot, F.; Canale, V.; Chaumont-Dubel, S.; Walczak, M.; Satała, G.; Nosalska, K.; Mahoro, G.U.; Słoczyńska, K.; Piska, K.; et al. Neuropathic Pain-Alleviating Activity of Novel 5-HT6 Receptor Inverse Agonists Derived from 2-Aryl-1H-Pyrrole-3-Carboxamide. Bioorganic Chemistry 2021, 115, 105218, doi:10.1016/j.bioorg.2021.105218.
- Drop, M.; Canale, V.; Chaumont-Dubel, S.; Kurczab, R.; SataÅ‚a, G.; Bantreil, X.; Walczak, M.; Koczurkiewicz-Adamczyk, P.; Latacz, G.; Gwizdak, A.; et al. 2-Phenyl-1 H -Pyrrole-3-Carboxamide as a New Scaffold for Developing 5-HT 6 Receptor Inverse Agonists with Cognition-Enhancing Activity. ACS Chem. Neurosci. 2021, 12, 1228–1240, doi:10.1021/acschemneuro.1c00061.
